# The Need for Oral Hygiene Care and Periodontal Status among Hospitalized Gastric Cancer Patients

**DOI:** 10.3390/jpm12050684

**Published:** 2022-04-26

**Authors:** Flavia Mirela Nicolae, Francesco Bennardo, Selene Barone, Petra Șurlin, Dorin Nicolae Gheorghe, Daniela Burtea, Ștefan Pătrascu, Sandu Râmboiu, Adrian Petru Radu, Bogdan Silviu Ungureanu, Adina Turcu-Știolica, Andreea Cristiana Didilescu, Victor Dan Eugen Strâmbu, Valeriu Marin Șurlin, Dan Ionuț Gheonea

**Affiliations:** 1Department of Periodontology, University of Medicine and Pharmacy of Craiova, 200349 Craiova, Romania; flavia.nicolae23@yahoo.com (F.M.N.); dorinngheorghe@gmail.com (D.N.G.); 2School of Dentistry, Department of Health Sciences, Magna Graecia University of Catanzaro, Viale Europa, 88100 Catanzaro, Italy; selene.barone@studenti.unicz.it or; 3Department of Gastroenterology, University of Medicine and Pharmacy of Craiova, 200349 Craiova, Romania; danaburtea26@gmail.com (D.B.); boboungureanu@gmail.com (B.S.U.); digheonea@gmail.com (D.I.G.); 4Department 1st of Surgery, University of Medicine and Pharmacy of Craiova, 200349 Craiova, Romania; stef.patrascu@gmail.com (Ș.P.); sandu_r@yahoo.com (S.R.); vsurlin@gmail.com (V.M.Ș.); 5Department of Surgery, Carol Davila University of Medicine and Pharmacy, 020021 Bucharest, Romania; drradupetru@yahoo.com (A.P.R.); victor.strambu@umfcd.ro (V.D.E.S.); 6Department of Pharmacoeconomics, University of Medicine and Pharmacy of Craiova, 200349 Craiova, Romania; adina.turcu@gmail.com; 7Department of Embryology, “Carol Davilla” University of Medicine and Pharmacy, 050474 Bucharest, Romania; andreea.didilescu@umfcd.ro

**Keywords:** oral hygiene, oral health, periodontal disease, gastric cancer

## Abstract

Poor oral hygiene leads to the accumulation of dental plaque, thus contributing to the initiation of periodontal disease (PD). Local infections can lead to systemic inflammatory responses, which are essential mediators for the evolution of systemic conditions or cancer tumorigenesis. Often, patients hospitalized with life-threatening and incapacitating disorders such as gastric cancer (GC) might lose interest in keeping their mouth healthy. This study evaluates oral hygiene, periodontal status, and the need for oral care and medical personnel to assist in achieving it in patients hospitalized with GC. This study was carried out on 25 patients with a diagnosis of GC, divided into two groups (GP—14 patients from the Gastroenterology Department, and SP—11 patients from the 1st Department of Surgery). Patients were examined on the day of admission (T0), the day of the medical procedure of endoscopy or surgery (T1), and the day of discharge (T2), recording the number of absent teeth, dental plaque (PI), bleeding on probing (BOP), probing depths (PPD), frequency of toothbrushing, and if the oral hygiene had been self-performed or assisted. Data were subjected to statistical analysis. Our results showed that, in both the GP and the SP group, there were strong and statistically significant correlations between PI and BOP measured on the last day of hospitalization and the period of hospitalization after the medical procedure. Longer hospital stays and the presence of surgery were risk factors for changing toothbrushing frequency. Results also highlight the need for a dentist to diagnose and eventually treat periodontal disease before and after hospitalization, and for a trained nurse who should help take care of the patient’s oral hygiene during hospitalization.

## 1. Introduction

Poor oral hygiene can lead to the build-up of high levels of dental plaque, which is a biofilm that hosts different microorganisms that can contribute to the initiation of periodontal disease. Periodontal disease (PD) is one of the most prevalent chronic diseases, characterized by a polymicrobial infection. Untreated, it can lead to severe bone and tooth loss caused by the production of proinflammatory cytokines that are upregulated by the involved bacteria [1,2]. Thus, the local infection can lead to systemic inflammatory responses [3], which are essential mediators for the evolution of systemic conditions or for gastric cancer tumorigenesis [4,5].

Numerous studies indicated an association between periodontal changes and systemic diseases. Periodontal infection influences short-term glycemic control in diabetic patients [6], with certain correlations being highlighted between periodontal disease and coronary heart disease, stroke, autoimmune and respiratory disorders [7]. Mature bacterial plaque can promote the growth of opportunistic respiratory pathogens with consequences for hospital-acquired pneumonia [8]. A large body of evidence has emerged regarding gastrointestinal tract cancers, linking periodontal infection with colorectal, pancreatic, and gastric cancers [4,9,10,11,12,13].

Gastric cancer (GC) is the 5th most common type of cancer [14], with high rates of mortality, as the 5-year survival rate is below 25%, and it is ranked the third leading cause of cancer-related death [15]. Adenocarcinomas are the most common type of GC, accounting for nearly 95% of all cases [16]. There is a need for its early detection for higher chances of survival [17,18]. 

Recent studies show that there may be a link between GC and PD through the bacteria of the periodontal biofilm [4], and a bacterial signature was found in gastric cancer tissue, with a higher abundance of *Fusobacterium*, which is one of the most important periopathogens [19]. In gastrointestinal tract cancers, oral *Fusobacterium* can travel to the cancerous tissue [20], worsening evolution and prognosis [21]. 

Oral health is a significant and basic part of general health, so it is vital to conserve it [22,23]. Older longitudinal research revealed that older adults who never brushed their teeth at night had a 20–35% increased mortality risk in comparison with those that brushed every night, thus stating that evening oral hygiene might be a significant prognostic factor of mortality [24]. Lee stated that gastrointestinal cancer was discovered less often in patients with good oral hygiene, and that correct oral habits could minimize the risk of gastroenteric cancer [25]. 

It is estimated that 44% to 65% of hospitalized care-dependent patients do not receive suitable oral care [26]. In terms of home care, another study stated that more than half of the caregivers were helping patients with nutrition, taking medication, or changing clothes, but just 26% helped the patients with oral hygiene [27,28]. A recent paper showed that patients who received oral care before the resection surgery of gastric neoplastic tissue had diminished risk of developing postoperative complications [29].

Often, patients hospitalized with life-threatening and incapacitating disorders such as gastric cancer might lose interest in keeping their oral cavity healthy, so healthcare workers must intervene. A dentist should offer guidance about maintaining oral health, but it is the nurse who should help the patient with daily dental hygiene, reducing the need for further complicated treatments.

There is a need for oral care both before and during hospitalization, when the patient should be helped by a trained nurse and a specialized dentist. Considering that oral hygiene is important in patients with digestive tract cancers, and that the scientific literature shows that periopathogens from dental biofilm may be involved in the evolution of gastroenterological cancers, the purpose of our study was to focus on gastric cancer patients, as GC is still quite widespread in Romania. 

### Aim

The aim of this study was to assess oral hygiene and periodontal status in patients hospitalized with gastric cancer, and the need for oral care and medical personnel to assist in achieving it.

## 2. Materials and Methods

### 2.1. Study Design

This was a prospective follow-up study conducted on hospitalized patients with a diagnosis of gastric adenocarcinoma, admitted to the County Hospital of Emergency of Craiova, University of Medicine and Pharmacy of Craiova between December 2020 and June 2021. 

### 2.2. Participants

At baseline, we examined a total of 33 patients hospitalized in the Gastroenterology Department (19 patients) and the 1st Department of Surgery (14 patients) of the Clinical County Hospital of Emergencies of Craiova, University of Medicine and Pharmacy of Craiova, suspected of having gastric cancer. During hospitalization, 4 of the patients died after the intervention (2 patients from the Gastroenterology Department and 2 patients from the 1st Department of Surgery). They could not be examined on the day of discharge, so they were excluded from this study. Participants were enrolled in this study if they had received a gastric adenocarcinoma diagnosis. Lastly, the sample in our study was represented by 25 participants that had received a diagnosis of gastric adenocarcinoma (14 patients from the Department of Gastroenterology and 11 from the 1st Department of Surgery) and who met the eligibility criteria. Inclusion criteria were: basic understanding of instructions and if the time between the first day of hospitalization and the day of the intervention was a minimum of 3 days. Exclusion criteria were: patients unwilling or incapable to sign the informed consent form, absence from any visit, unconsciousness, or if the oral examination could not be performed. 

The study was initiated after obtaining approval from the ethics committee. All participants agreed to sign an informed consent form before participating in the study. All procedures were approved by the Research Ethics Commission of the University of Medicine and Pharmacy of Craiova, no. 127/09.12.2019, and the Clinical County Hospital of Emergency of Craiova, no. 3273/21.01.2020, and were performed in accordance with the guidelines of the Declaration of Helsinki.

Enrolled patients were recruited regardless of age, gender, or cancer stage. Medical charts were used to obtain appropriate diagnostic information. Participants were divided into two groups depending on the department in which they were hospitalized given that they had different hospitalization periods, different interventions, either upper endoscopic examinations or surgical removal of gastric cancer, and different invalidations, especially patients from the 1st Department of Surgery, which required different care. Patients from the 1st Department of Surgery were included in the SP group, and patients from the Gastroenterology Department were included in the GP group. 

### 2.3. Clinical Assesments

All clinical assessments were performed by a single experienced examiner. Examinations were conducted in the morning, after the patients had eaten, and oral hygiene was performed. 

On the first day of hospitalization (T0), we recorded any experienced difficulties when eating (yes or no), and if the patient had secondary anemia (yes or no) [30]. Patients received specific recommendations on how to correctly brush their teeth. Periodontal examination was performed with the UNC15 periodontal probe (Hu-Friedy, Chicago, IL, USA) recording the number of absent teeth (AT), plaque index (PI), bleeding on probing (BOP), and periodontal probing depth (PPD) for all teeth, which were recorded in the periodontal chart. 

The number of absent teeth (AT) was recorded in the patient’s periodontal chart, excluding residual tooth roots and 3rd molars. Periodontal probing depth (PPD) was examined in 6 sites (mesiobuccal, centrobuccal, distobuccal, mesio-oral, centro-oral, disto-oral) for every tooth present, and was recorded in the periodontal chart of the patient. PPD for each patient was obtained by summing those values and dividing them by the number of examined sites. For the PI assessment, every tooth was examined in the same 6 sites as for the PPD, and the presence or absence of PI was recorded for every site. The number of sites where PI was present was summed, divided by the number of total sites examined, and multiplied by 100. The same method was used for BOP assessment. Lastly, dental plaque and the number of bleeding sites were recorded as a percentage. 

The morning before the intervention (T1) and the last day of hospitalization (T2), the oral examination was reperformed, and AT, PI, BOP, and PPD assessments were repeated.

At each of the three moments of examination (T0, T1, T2), patients were motivated to maintain appropriate oral hygiene, and the self-reported frequency of toothbrushing was recorded (less than daily, once a day, once to twice a day, or more than twice a day) and whether the oral hygiene was self-performed or performed with assistance. 

### 2.4. Statistical Analysis

Data expressed as mean, standard deviation (SD), median, and interquartile range were subjected to statistical analysis in order to detect differences between subgroups (time periods or patient groups) using the Mann–Whitney test. The existence of statistical correlations between the different datasets using Spearman’s coefficients was assessed. 

Significant independent predictors for toothbrushing change were analyzed using univariate and multivariate analysis in a logistic regression (the binary-dependent variable was toothbrushing with “yes” and “no” values). The 95% confidence interval for the odds ratio was evaluated for every predictor. The area under the receiver operating characteristic (AUC) curve and its 95% CI were assessed to demonstrate the accuracy of the proposed model. All data analyses were performed using GraphPad Prism 9.3.1 (GraphPad Software, San Diego, CA, USA). A *p*-value of less than 0.05 was considered to be statistically significant. 

## 3. Results

### 3.1. Patient Data

The age of the 25 eligible patients ranged from 53 to 85 years, and the median age was 70.24 years (SD = 10.24). The study sample was almost equally distributed by gender: 14 patients were males (56%), and 11 were females (44%). 

Anemia was present in 20 patients (80%) (78.57% of the GP, 81.81% of SP), and 16 patients (64%) reported difficulties when eating (57.14% of GP, 72.72% of SP). All 25 (100%) patients stated at T0, T1, and T2 that toothbrushing was self-performed at home and during hospitalization. Data about the patients are shown in Table 1. 

### 3.2. Hospitalization Periods

Patients remained hospitalized for an average of 14.88 days (SD = 12.01). Patients from the Gastroenterology Department were hospitalized for up to 13 days, with a mean hospital stay of 8.14 days (SD = 2.90). Patients from the 1st Department of Surgery were hospitalized for up to 60 days, with a mean average stay of 23.45 days (SD = 13.83). 

The average period prior to the intervention was 5.76 days (SD = 3.84). For GP, the average value was 3.78 days (SD = 1.18); for SP, it was about 8.27 days (SD = 4.60). Mean hospital stay after intervention was about 9.12 days (SD = 9.79). For GP, the time was 4.35 days (SD = 2.39); for SP, it was 15.18 days (SD = 12.31). 

### 3.3. Periodontal Parameters

All results concerning periodontal parameters are described in Table 2. The average number of AT was 17.68 (SD = 7.40). AT was not modified over the hospitalization period. The mean value for PI at T0 was 65.44% (SD = 8.51%); at T1, it was 72.36% (SD = 10.62%); and at T2, it was 88.52% (SD = 10.14%). The average BOP at T0 was 61.32% (SD = 10.90%); at T1, it was 67.04% (SD = 11.56%); and at T2, it was 76.24% (SD = 14.33%). Mean PPD at T0 was 5.28 mms (SD = 1.85 mms); at T1, it was 5.28 mms (SD = 1.86 mms); and at T2, it was 5.30 mms (SD = 1.87 mms).

### 3.4. Statistical Differences between Assesed Parameters

Statistically significant differences were found between the total period of hospitalization (T0–T2) in GP and SP (*p* < 0.001). Statistically significant differences were discovered regarding the two periods of hospitalization (T0–T1 and T1–T2) between the two groups (GP and SP) (*p* < 0.01, respectively *p* < 0.001). 

Tests for statistical differences were applied within each group for the evaluated parameters (PI, BOP, PPD) between the three moments of the study (T0, T1, T2). The same tests were also applied between the 2 groups for the same parameters (PI, BOP, PPD) at each moment of the study (T0, T1, T2). 

Statistical differences (*p* < 0.05) were assessed between the values of some parameters (PI and BOP), at T0, T1, T2 between SP and GP groups (Table 3), and in every group, as shown in Table 4. No statistical differences were found in GP between the analyzed parameters (PI, PPD, BOP) at T0 and T1, or between GP and SP for the same parameters evaluated at T0. No statistical difference concerning PPD was observed between GP and SP groups at T1 and T2, as shown in Table 3 and Table 4. 

### 3.5. Correlations between Periodontal Indices and Hospitalization Periods

Strong and statistically significant correlations were found between T0–T2 and the presence of the surgical intervention (r = 0.85, *p* < 0.01).

Weak and statistically not significant correlations between PI at T1 and T0–T1 (r = 0.34, *p* > 0.05) and between BOP at T1 and T0–T1 (r = 0.28, *p* > 0.05) were found in GP, whereas the same correlations of PI and BOP in SP were moderate and statistically not significant (r = 0.57, *p* > 0.05; r = 0.50, *p* > 0.05).

In both GP and SP, strong and statistically significant correlations were found between PI at T2 and T1–T2 (r = 0.79, *p* < 0.001, for SP, and r = 0.73, *p* < 0.01, for GP) and between BOP at T2 and T1–T2 (r = 0.68, *p* < 0.001, for SP, and r = 0.64, *p* < 0.01, for GP). 

### 3.6. Oral Hygiene Habits

Oral hygiene habits are described in Table 5.

Out of the examined patients, 13 participants did not use the toothbrush after the intervention, and 5 of them totally ceased oral hygiene since being hospitalized. From the 13 participants that did not perform oral hygiene after the intervention, 8 were SP and 5 were GP.

In multivariate analysis, simple logistic regression was first applied to assess the risk of toothbrushing behavior changing in relation to surgery (yes/no) and total hospital days T0–T2. Both were included in multivariate analysis and showed that longer hospital stays (OR = 2.68; 95% CI, 1.21–14.40) and the presence of surgery (OR = 0.0004511; 95% CI, 2.9 × 10^−10^–0.52) were risk factors for changes in toothbrushing frequency. The accuracy of the model evaluated by AUC, as shown in Figure 1, was very high, at 0.93 (95% CI, 0.83–1.00, *p*-value = 0.0172) for the T0–T2 period. 

In the regression approach of assessing the risk of toothbrushing behavior changing in relation to surgery (yes/no) and T1–T2 hospital days, longer hospital stays (OR = 2.25; 95% CI, 1.03–12.88) and surgery (OR = 0.0096; 95% CI, 2.4 × 10^−7^–3.029) were also risk factors (AUC = 0.85, 95% CI, 0.69–1, *p*-value = 0.0544). In univariate analysis, crude odds ratio (1.96, 95% CI, 0.82–11.30) and area under the ROC curve (0.79, *p*-value = 0.20) demonstrated no correlation between toothbrushing behavior changing and T1–T2 hospital days for patients in gastroenterology without surgery. 

## 4. Discussion

Considering that the aim of this study was to evaluate oral hygiene and periodontal status in patients hospitalized with gastric cancer, and the need for oral care and medical personnel to help with it, our findings revealed that SP patients had higher PI and BOP values than those of GP patients at any examination time, and that these higher values were related to an increased period of hospitalization and a decline in toothbrushing frequency. 

In our study, toothbrushing frequency decreased from the first day of hospitalization to the day of intervention, but the greatest decline was seen in SP, where more than half of the patients ceased brushing after intervention. Longer hospital stays and the presence of surgical intervention were also risk variables for changing toothbrushing frequency throughout the hospitalization period. Furthermore, in the present study, longer hospitalization periods following surgical intervention were risk factors for changing oral hygiene frequency, whereas there was no correlation between the time after the endoscopic examination and the frequency of toothbrushing in patients who did not undergo surgical intervention.

The cessation of oral hygiene for more than 72 h enables the accumulation of periopathogens from the oral biofilm, which could lead to important changes in the immunoinflammatory status of the patient and to the initial lesion of gingival inflammation. If oral habits are not re-established, this evolves into an early, advanced, and stable lesion in which periodontal pockets are present [31]. Microorganisms from the subgingival biofilm can spread to other tissue or structures, and, especially in hospitalized patients with impaired immunity or those subjected to surgeries, bacteria are associated with the degeneration of systemic health [32,33,34]. The surgical wound could be colonized by oral bacteria, swallowed or travelling through a hematogenous route, which may lead to bacteriemia, which is highly correlated with mortality in postsurgical patients [35]. Increasing evidence suggests that the bacterial infection of the oral cavity can cause chronic inflammation, which further contributes to the development or advancement of various types of cancer [3,36,37]. 

Multiple studies reported strong correlations between toothbrushing frequency and metabolic syndrome, hemodialysis [38,39,40], or cerebral strokes [5]. The incidence of hospital-acquired pneumonia, which is frequently associated with plaque or oral debris [8], declined in nonventilated patients who performed oral hygiene [41]. The prevalence of cancer, hypertension, and diabetes was lower in patients who brushed their teeth more than once a day [30], while toothbrushing once or less than once a day was significantly associated with an increased risk for gastric cancer, having a 5.6 higher risk than that of the controls [25,42].

In comparison with other types of malignancies, GC is one of the most aggressive types of cancer, often followed by anemia, and most of the patients reported difficulties eating and a decrease in their quality of life, which can be considered to be prognostic factors for a worse prognosis of GC [43]. In our study, 76% of the patients presented secondary anemia, and 64% reported that they had difficulties when eating. This could be attributed to the evolution of GC, but also because of the discomfort from the oral cavity given the periodontal status of the patient with increased bleeding in association with poor oral hygiene. Considering that the average PPD was higher than 5 mm and this can often be accompanied by tooth mobility, bad taste, and smell, all these elements impair correct alimentation. 

Loss of teeth and oral health issues were positively correlated with gastric cancer [25,44], and meta-analysis concluded that patients with edentulism have a higher incidence of gastric cancer [45]. Our patients lost approximately 17.68 teeth, even more than that in a previous study that showed that patients with digestive cancer had approximately 12 absent teeth [46], probably leading to inadequate mastication. It was postulated that insufficient mastication and a lower amount of saliva because of fewer teeth put an extreme digestive burden on the affected stomach, further leading to the development of gastric neoplasia [9]. 

Carrilho et al. showed that the majority of hospitalized patients had poor oral hygiene, increased rates of edentulousness, and periodontal disease [33]. He also reported gingival bleeding in 94.5% and gingival inflammation in 98.1% of the examined patients, reflecting a poor oral health status in hospitalized patients and stated that 50% of them could not visit the dental facility of the hospital, nor could they perform correct daily dental hygiene [33]. Our aim was not to diagnose periodontal disease, but to emphasize the need for oral care, which results from changes in PI and BOP during hospitalization. Gingival inflammation and dental plaque abundance were positively associated with increased time of hospitalization, so oral hygiene care must be performed even more in patients hospitalized over a longer period to reduce plaque abundance and consequently lower the number of periopathogens [33]. In our study, we found strong correlations between the total period of hospitalization and the presence of surgery. This could be explained by the fact that patients who had undergone surgical intervention requiring more complex pre- and postoperative care. Prior to intervention, SP patients remained hospitalized longer than GP patients did because of the time needed for more investigations or because they had reached the hospital with a higher number of complications to be investigated than GP patients did. After intervention, SP remained hospitalized for approximately 3 times longer than GP patients were. This difference could be related to the two types of interventions, with the surgical one being more complex and invalidating, so more time is needed for the patients’ recovery. 

In our study, the frequency of toothbrushing declined from the first day of hospitalization to the day of intervention, but the highest decline was observed in SP, where more than half of the patients completely stopped oral hygiene after the intervention. Our analysis also showed that longer hospital stays and the presence of surgical intervention were risk factors for changing the toothbrushing frequency during the entire period of hospitalization. Moreover, the longer hospitalization time after the surgical intervention was a risk factor for changing the frequency of oral hygiene, while for patients that had not had a surgical intervention, we found no correlation between time after endoscopic examination and frequency of toothbrushing. We should consider that, for SP, surgical intervention was more invalidating, thus impairing their general mobility in comparison with GP. Patients that had been subjected to surgical intervention are often physically incapacitated after the surgery, or they may be psychologically debilitated as they think more about the disease and its consequences, thus becoming demotivated and less preoccupied with personal hygiene. This is closely related to the collapse of PI and the heightening of BOP in relation to the decreased frequency of toothbrushing. 

Srinivasan reported that 93% of hospitalized elderly patients had their own toothbrushes, and that 72% brushed their teeth at least once a day since hospitalization, although these data need to be interpreted cautiously as they are reported by the patients themselves which could not be accurate [47]. In another paper, participants stated that their oral hygiene had worsened since hospitalization, as they lacked the motivation or were limited by physical barriers for toothbrushing, and they were open to being assisted with oral care [48]. Another study revealed that some elderly patients had given up on oral hygiene because they never thought about asking for help from a nurse, while others wanted to brush their teeth for as long as they could by themselves [49]. 

Damaged teeth and poor oral hygiene were more frequent in patients who suffered from gastric cancer than they were in the control group [50]. On the day of presentation to the hospital, our patients showed a mean PI of about 65%, BOP of approximately 61%, and the average value for PPD was 5.28 mm, which were comparable to an older study that revealed that more than half of GC patients presented BOP, clinical attachment loss higher than 3 mm, PPD higher than 6 mm, and severe or moderate forms of gingival inflammation [51]. Considering these values with a PPD greater than 5 mm, which could mean deep periodontal damage, even though we did not evaluate the clinical attachment loss, the suspicion of periodontitis appears, and thus the need for specialized medical personnel to diagnose and treat PD both before and after hospitalization arises. Even though we did not establish periodontal diagnosis for our patients, as it was not our aim, we emphasize the need for oral care that results from the fall of PI and BOP. At that time, BOP for our patients ranged from 39% to 79%, so all of our patients presented with a minimum of gingivitis, according to the latest classification of periodontal disease [52]. Considering that these values were found on the first day of hospitalization, we could assume that even before admission, patients had poor periodontal status and low oral hygiene. Hence, the need to be supervised and treated by a dentist or periodontist arises, either before hospital admission, or after discharge.

Our study showed that SP patients had higher values than those of GP of both PI and BOP at any examination time, and were associated with an increased number of hospitalization days and a decrease in toothbrushing frequency. This could be explained by the fact that they are more preoccupied of or affected by the general illness, so their motivation for personal hygiene would decrease. As the number of days of hospitalization increases, so do PI and BOP. PI is the expression of oral hygiene, while BOP may point to the inflammatory status, both of them suffering modifications over a longer period of time. The PI in the GP group did not change before the endoscopic intervention, as these patients have shorter hospitalization periods, while in the SP group, PI changes, with surgical patients being hospitalized longer before the intervention. 

The lack of correlation between PI and BOP in the morning of the intervention and the period prior to the intervention in both groups could be explained by the fact that the period was too short for these parameters to change, as a minimal period of 72 h is needed for the initial lesion to occur [31]. On the other hand, strong correlations between PI and BOP in the morning of discharge and the period after the intervention in both groups could be explained by the fact that, after the intervention, the patients might be physically or psychologically incapacitated, thinking about their general illness and might neglect oral hygiene, which then leads to the accumulation of oral plaque and consequently to heightening BOP. 

Our results highlighted that levels of both PI and BOP increased from the first to the last day of hospitalization, while the frequency of toothbrushing decreased. These parameters were correlated with increased periods of hospitalization. Considering that patients stated that toothbrushing was self-performed, our results sustain the idea that patients need help in performing oral hygiene during hospitalization. Moreover, we divided the patients from the start into two groups on the basis of medical intervention that invalidated them more or less. The attention of the staff regarding oral care should be adapted to the specifics of each department. 

All of our patients reported that oral hygiene was self-performed, but taking into account the values of PI, BOP, and PPD, there is a need for specialized healthcare workers during hospitalization arises, and for a dentist or periodontist to diagnose and eventually treat periodontal disease both before and after hospitalization [53]. Patients often do not know how to correctly eliminate dental plaque from their teeth, tongue, or gums, and cannot remove the oral biofilm from periodontal pockets, which remain permanent reservoirs of bacteria, so the contribution of a dental specialist is required. Moreover, all patients should have access to professional dental care, which can be completed by the nurses afterwards [28]. Certain studies revealed that nurses helped patients regarding oral hygiene in just 36% of the cases, and did not intervene if they were not asked for help, but if asked, they would gladly help the patients with oral hygiene [54,55,56]. A possible cause for not providing oral cleanliness could be a lack of time, as there is a continuous need for prioritizing tasks, especially the administration of medications, which consumes most of the time [28,57]. Moreover, caregivers were worried about being perceived as disrespectful, and they were hesitant to intervene if not requested for help [48], as another study reported that approximately 50% of the patients declined the offer of assistance from the healthcare workers [28]. Nurses need more information about the systemic health benefits of oral care, and the skills to be able to perform adequate oral hygiene [28]. Gibney revealed a 35% respectively 37% improvement in oral hygiene when dental hygienists or medical assistants, who were previously trained, supervised and helped patients with oral care [58]. 

A nationwide retrospective observational study from Japan, with more than 500,000 participants, showed that prior to major cancer surgeries, only 16% of the participants received oral care from a dental specialist. A statistically significant association was found between preoperative oral care and a decrease in postoperative pneumonia and all-cause mortality in the first 30 days after surgery [59]. Moreover, before resection surgery for gastric cancer, patients that received intensive oral care prior to the intervention had a decreased risk of developing postoperative infectious complications [29]. However, if nurses receive education and training in order to successfully implement oral hygiene procedures, this could result in a healthier oral status, which translates in fewer cases of hospital-acquired pneumonia [58]. 

There is a need for dental professionals in every hospital, for the improvement of oral health problems, which could lead to the enhancement, with some limitations, of the general conditions and the improvement of the way patients feed themselves. Moreover, a research paper emphasized that for the primary prevention of gastric cancer, eradicating periodontal infections and stabilizing periodontal disease through periodontal treatment should be considered [4]. Hwang stated that, in patients with PD who had received professional periodontal treatment, the risk of developing cancer drastically declined in comparison with the control group [60]. Patients should constantly be reminded of oral hygiene, and it should be emphasized that oral care is a mean of prevention of future diseases [28]. Maintaining oral health allows for appropriate mastication with a complete and nutritional diet, a good esthetic aspect, with implications in personal image and self-esteem, and, in the absence of oral pain, his overall life quality may be improved. It is imperious to consider more discussions about oral health and hygiene in the field of nursing, as these health workers are directly implicated in patients’ hygiene and could strongly impact patients’ oral health [28,57]. There is also a need for the elaboration of an oral care guideline based on multidisciplinary approaches and research evidence [61]. 

The study’s main limitation was the small number of participating patients, generated by the fact that the study was implemented during the COVID-19 pandemic, when the County Hospital of Emergency of Craiova was declared a support hospital for COVID-19 infections for a very long time period. This significantly impacted the study’s ability to gather a larger sample of patients. Besides the restrictions imposed during the peak periods of the pandemic, there was also a reduced addressability of patients towards treatment of other diseases. Considering these aspects, we intend to expand the study to a larger sample of patients when the prepandemic situation returns. 

Regarding the modality by which oral hygiene was performed, self-performed or performed with assistance, our study has some limitations. Considering that all patients stated at every time of the examination that toothbrushing was self-performed and that the frequency of performing oral hygiene has decreased, we did not investigate the reasons behind the nurses’ absence from assistance. We also did not interview the nurses if they were trained to help patients with toothbrushing, if they had time or if they offered and were refused. This latter aspect is intended to be detailed in further research in order for nurses to be as trained and involved as possible in improving the patients’ life quality. 

Given that the purpose of this present study was to assess the need for oral hygiene care, our results showed that there is a great need for this in hospitalized GC patients. As toothbrushing habits were self-performed by the patients, we could assume that, regardless of how well we motivated them during each of the three examination moments, it was not enough for them to maintain appropriate oral hygiene, so the help of a trained healthcare worker is needed. Depending on each health and education system, the hospital unit, and specific department, protocols could be developed to maintain the frequency of oral hygiene and for designating healthcare workers to assist. Our results are based on self-reported data by the patients (frequency and modality of toothbrushing) and are prone to subjectivism. There should be specialized protocols and charts to be filled in by the nurses regarding the maintenance of patients’ oral hygiene, so that subjectivism could be eliminated. 

## 5. Conclusions

Within the limitations of our study, results highlighted the need for oral care before, during, and after hospitalization by qualified medical personnel, eventually diagnosing and treating PD in patients with GC.

## Figures and Tables

**Figure 1 jpm-12-00684-f001:**
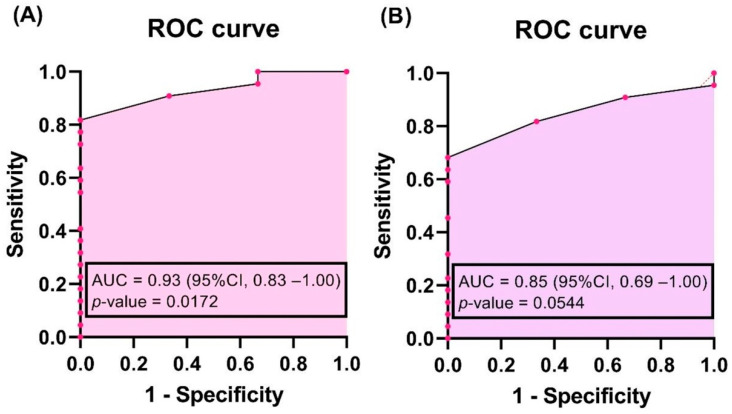
Receiver operating characteristic curve for modelling toothbrushing presence. (**A**) T0–T2; (**B**) T1–T2.

**Table 1 jpm-12-00684-t001:** Patient data.

		GP		SP	
		*n*	%	*n*	%
**Gender**	**M**	8	57.14	6	54.54
	**F**	6	42.85	5	45.45
**Anemia**		11	78.57	9	81.81
**Difficulty when eating**		8	57.14	8	72.72
**Self-performed toothbrushing**		14	100	11	100

*n*, number of patients in each group; %, percentage of patients in each group.

**Table 2 jpm-12-00684-t002:** Periodontal parameters (mean ± standard deviation, median, interquartile range, range).

Time of the Examination	Department	AT (*n*)	PI (%)	BOP (%)	PPD (mms)
T0	GP	21 ± 6.71 22, 16.5–25.25 7–30	62.92 ± 10.41 63.5, 55.75–70.25 45–83	54.28 ± 8.66 56, 47.25–61.25 39–67	5.02 ± 1.77 5.15, 4.3–6.4 1.8–7.7
	SP	13.45 ± 6.15 16, 6–18 3–20	68.63.63 ± 3.55 69, 65–71 63–74	70.27 ± 5.51 69, 65–75 64–79	5.60 ± 1.99 6.3, 5–7 1.8–7.8
T1	GP	21 ± 6.71 22, 16.5–25.25 7–30	67.42 ± 11.39 69, 54.75–77.25 52–88	58.85 ± 7.80 61, 51.5–65.25 45–69	5.02 ± 1.77 5.15, 4.3–6.4 1.8–7.7
	SP	13.45 ± 6.15 16, 6–18 3–20	78.63 ± 4.94 78, 74–83 71–86	83.45 ± 5.37 77, 72–83 70–86	5.60 ± 2.00 6.3, 5–7 1.8–7.9
T2	GP	21 ± 6.71 22, 16.5–25.25 7–30	83.35 ± 10.51 85.5, 71.75–92.75 69–98	66.5 ± 10.50 67, 55.75–74.5 50–82	5.04 ± 1.77 5.15, 4.3–6.4 1.8–7.7
	SP	13.45 ± 6.15 16, 6–18 3–20	95.09 ± 4.32 91, 94–100 90–100	77.45 ± 5.37 86, 83–96 83–100	5.64 ± 2.02 6.3, 5–7.1 1.8–7.9

GP, gastroenterology patients; SP, surgery 1st patients; AT, absent teeth; PI, Plaque Index; BOP, bleeding on probing; PPD, periodontal probing depth; T0, examination upon first day of hospitalization; T1, examination in the morning of the intervention; T2, examination on the last day of hospitalization.

**Table 3 jpm-12-00684-t003:** Differences among moments of examination regarding PI, BOP, PPD, in GP and SP (*p*).

	T0	T1	T2
	GP/SP		
	**PI** **BOP** **PPD**		
**T0**	-	0.29/<0.0001 0.15/<0.01 1/1	<0.0001/<0.0001 <0.01/<0.0001 0.98/0.92
**T1**	0.29/<0.0001 0.15/<0.01 1/1	-	<0.0001/<0.0001 0.04/<0.001 0.98/0.93
**T2**	<0.0001/<0.0001 <0.01/<0.0001 0.98/0.92	<0.0001/<0.0001 0.04/<0.001 0.98/0.93	-

T0, examination upon the first day of hospitalization; T1, examination in the morning of the intervention; T2, examination on the last day of hospitalization; *p* < 0.05 for statistically significance.

**Table 4 jpm-12-00684-t004:** Differences between groups GP and SP at T0, T1, and T2 regarding PI, BOP, PPD *(p*).

GP/SP	T0	T1	T2
	**PI** **BOP** **PPD**		
**T0**	0.07 0.08 0.20	-	-
**T1**	-	<0.01 <0.0001 0.20	-
**T2**	-	-	<0.01 <0.0001 0.18

T0, examination upon the first day of hospitalization; T1, examination in the morning of the intervention; T2, examination on the last day of hospitalization; *p* < 0.05 for statistically significance.

**Table 5 jpm-12-00684-t005:** Oral hygiene habits.

	Frequency of Toothbrushing	GP	SP
T0	More than twice a day	14.28%	9.09%
	Once to twice a day	42.85%	36.36%
	Once a day	42.85%	54.54%
	Less than daily	0%	0%
T1	More than twice a day	0%	0%
	Once to twice a day	28.57%	27.27%
	Once a day	57.14%	45.45%
	Less than daily	14.28%	27.27%
T2	More than twice a day	0%	0%
	Once to twice a day	14.28%	9.09%
	Once a day	50%	18.18%
	Less than daily	35.71%	72.72%

T0, examination upon the first day of hospitalization; T1, examination in the morning of the intervention; T2, examination on the last day of hospitalization.

## Data Availability

Data used to support the findings of this study are available from the corresponding author upon reasonable request.

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
