# Peer review of "The Need for Oral Hygiene Care and Periodontal Status among Hospitalized Gastric Cancer Patients"

_jpm, 2022, doi:10.3390/jpm12050684_

Round 1

Reviewer 1 Report

The authors presented one very significant one that refers to patients with gastric cancer. The importance of oral hygiene is of particular importance in hospitalized patients.
The study provides very useful information and is based on a very good methodology.

The introduction and discussion are too long, so I suggest they be revised. In the introduction, it is necessary to eliminate passages that are generally known and which refer to oral health.
Include the most significant results of your work in the discussion first, and then discuss similar studies.

Author Response

Thank you for reviewing our paper.

The authors presented one very significant one that refers to patients with gastric cancer. The importance of oral hygiene is of particular importance in hospitalized patients.
The study provides very useful information and is based on a very good methodology.

  • The introduction and discussion are too long, so I suggest they be revised. In the introduction, it is necessary to eliminate passages that are generally known and which refer to oral health.

Thank you for your valuable suggestion. As other reviewer also suggested, the introduction and discussion have been revised and reorganized, removing dispensable background information and synthetizing the conclusions of other studies included in the discussions.

  • Include the most significant results of your work in the discussion first, and then discuss similar studies.

Done. The discussion section has been reorganized, following the reviewer’s suggestions.  

Reviewer 2 Report

I have no comments or suggestions for Authors.

Author Response

Thank you for reviewing our paper.

Reviewer 3 Report

This study addresses oral hygiene practice and periodontal status among gastric cancer patients. While the topic is interesting, many points should be addressed to enhance the quality of the manuscript.

General comment:

The manuscript has many language errors and should be proofread.

Abstract:

  • Add the statistical methods used.
  • Add information on the number of participants.
  • “Our results 36highlighted that in both GP and SP groups there have been found strong and statistically significant correlations between PI and BOP at T2 and T1-T2” the sentence is not clear. Rewrite, please.
  • L93: you did not mention that toothbrushing frequency would be measured earlier.
  • The aim is not clear and should be rephrased.

Introduction :

  • While the introduction contains brief and useful information on the topic, the rationale for conducting the study and the gap in the literature is still not clear. Please add a paragraph that covers this.
  • The aim of the study is not clear. Rephrase, please.

Methods:

L111-116: this part is not needed here. Remove, please.

L126: “diagnostic” should be replaced by “diagnosis”.

L162: more information is needed regarding how dental plaque was examined and scored.

L166-169: how was this examined? Self-report? Elaborate, please.

L177: what is the type of the regression, give more details, please.

Results:

L183-190: it would be useful to add a table that contains the sample characteristics.

L202-202: what about other periodontal parameters. Describe the findings.

L208-222: The findings are poorly reported. Please describe and interpret the findings without emphasis on statistical significance.

Tables: report the number in 2 decimal places throughout the manuscript.

Discussion: start the discussion by recapping the aim of the study and the main findings of the study.

Author Response

Dear Reviewer, thank you for your time and effort to review our paper and for the valuable suggestions.

This study addresses oral hygiene practice and periodontal status among gastric cancer patients. While the topic is interesting, many points should be addressed to enhance the quality of the manuscript.

  • General comment:
    • The manuscript has many language errors and should be proofread.

The English language and vocabulary have been revised, anyway the Editor will do a full English revision for Authors that are not English native speakers.

  • Abstract:
    • Add the statistical methods used.  

Thank you for your valuable suggestion, as the statistical analysis was absent from the abstract. Because of the limiting number of words used in the abstract, we could not mention all the statistical methods used.

    • Add information on the number of participants.

Done, please see L32-33

    • “Our results 36highlighted that in both GP and SP groups there have been found strong and statistically significant correlations between PI and BOP at T2 and T1-T2” the sentence is not clear. Rewrite, please.

Done, please see L38-40

    • L93: you did not mention that toothbrushing frequency would be measured earlier.

Done

    • The aim is not clear and should be rephrased.

Done, please see L29-30

  • Introduction :
    • While the introduction contains brief and useful information on the topic, the rationale for conducting the study and the gap in the literature is still not clear. Please add a paragraph that covers this.

Done, please see L104-107

    • The aim of the study is not clear. Rephrase, please.

Done, please see L115-1117

  • Methods:
    • L111-116: this part is not needed here. Remove, please.

Done

  • L126: “diagnostic” should be replaced by “diagnosis”.

Done

  • L162: more information is needed regarding how dental plaque was examined and scored.

Done, please see L176-180

  • L166-169: how was this examined? Self-report? Elaborate, please.

Done, please see L185-187

  • L177: what is the type of the regression, give more details, please.

Done, please see L196-198

  • Results:
    • L183-190: it would be useful to add a table that contains the sample characteristics.

Done, Table 1 created

  • L202-202: what about other periodontal parameters. Describe the findings.

Done, please see L229-233

  • L208-222: The findings are poorly reported. Please describe and interpret the findings without emphasis on statistical significance.

For an improved comprehension and increased scientific accuracy, the results have been presented and described in accordance with the statistical analysis (2.4. Statistical analysis), using the statistical significance, as a way to reflect the relevance of our results. The interpretation of the findings without emphasis on the statistical significance has been included in the discussion section.

  • Tables: report the number in 2 decimal places throughout the manuscript.

Done

  • Discussion: start the discussion by recapping the aim of the study and the main findings of the study.

Done, please see L309-323.

Thank you in advance for your willingness to read the revised manuscript.

Round 2

Reviewer 3 Report

The authors successfully addressed my comments. The manuscript has improved. I have no further comments to add.